# Non-Coding RNAs: Prevention, Diagnosis, and Treatment in Myocardial Ischemia–Reperfusion Injury

**DOI:** 10.3390/ijms23052728

**Published:** 2022-03-01

**Authors:** Mihnea-Cosmin Marinescu, Andrada-Luciana Lazar, Monica Mihaela Marta, Angela Cozma, Cristina-Sorina Catana

**Affiliations:** 1County Clinical Emergency Hospital of Brasov Romania, 500326 Brașov, Romania; mihnea.marinescu@umfcluj.ro; 2Department of Vascular Surgery, Second Surgical Clinic, “Iuliu Haţieganu” University of Medicine and Pharmacy, 400012 Cluj-Napoca, Romania; 3Department of Dermatology, “Iuliu Hatieganu” University of Medicine and Pharmacy, 400012 Cluj-Napoca, Romania; andradalucianalazar@yahoo.com; 4Department of Medical Education, “Iuliu Hatieganu” University of Medicine and Pharmacy, 400012 Cluj-Napoca, Romania; mmarta@umfcluj.ro; 5Department of Internal Medicine, “Iuliu Haţieganu” University of Medicine and Pharmacy, 400012 Cluj-Napoca, Romania; 6Department of Medical Biochemistry, “Iuliu Haţieganu” University of Medicine and Pharmacy, 400012 Cluj-Napoca, Romania; ccatana@umfcluj.ro

**Keywords:** microRNA, long non-coding RNA, cardiovascular diseases, ischemia–reperfusion injury, biomarker, acute myocardial infarction

## Abstract

Recent knowledge concerning the role of non-coding RNAs (ncRNAs) in myocardial ischemia/reperfusion (I/R) injury provides new insight into their possible roles as specific biomarkers for early diagnosis, prognosis, and treatment. MicroRNAs (miRNAs) have fewer than 200 nucleotides, while long ncRNAs (lncRNAs) have more than 200 nucleotides. The three types of ncRNAs (miRNAs, lncRNAs, and circRNAs) act as signaling molecules strongly involved in cardiovascular disorders (CVD). I/R injury of the heart is the main CVD correlated with acute myocardial infarction (AMI), cardiac surgery, and transplantation. The expression levels of many ncRNAs and miRNAs are highly modified in the plasma of MI patients, and thus they have the potential to diagnose and treat MI. Cardiomyocyte and endothelial cell death is the major trigger for myocardial ischemia–reperfusion syndrome (MIRS). The cardioprotective effect of inflammasome activation in MIRS and the therapeutics targeting the reparative response could prevent progressive post-infarction heart failure. Moreover, the pharmacological and genetic modulation of these ncRNAs has the therapeutic potential to improve clinical outcomes in AMI patients.

## 1. Introduction

Myocardial ischemia–reperfusion (I/R) injury in acute myocardial infarction (AMI) is the most important cause of morbidity and mortality worldwide [1].

The exact mechanisms through which the homeostasis of myocardial cells is affected during I/R injury of the heart are not completely understood [2]. Pathological changes such as inflammation, autophagy, apoptosis, calcium overload, neurohumoral activation, and oxidative stress are considered to have the same underlying cause as I/R injury [3]. Therefore, new biomarkers for the prevention, monitoring, and treatment of I/R must be identified and the most important challenges of such an integrative therapeutic approach must be understood.

In the last decade, the development of high-throughput techniques in sequencing technology allowed a better understanding of the complexity of the human transcriptome by showing that the non-coding portion of the genome plays a more significant role in human pathology [4,5,6]. Therefore, the human genome is transcribed into various classes of functional non-coding RNAs that are powerful regulators of a multitude of cellular and pathological processes [7].

Based on their size, molecules have fewer than 200 nucleotides for short non-coding RNAs including microRNAs (miRNAs), while long ncRNAs (lncRNAs) have more than 200 nucleotides. Long ncRNAs can also present in a circular form, called circular RNAs (circRNAs). The three major types of ncRNAs (miRNAs, lncRNAs, and circRNAs) act as signaling molecules closely involved in many cardiovascular disorders. I/R involves the atypical modulation of mitochondrial function and the autophagic and apoptotic signaling pathways. More recently, non-coding RNAs, including long non-coding RNAs (lncRNAs) and microRNAs (miRNAs), have been shown to influence I/R injury. The epigenetic regulation of miRNAs and lncRNAs also plays a key role in IR injury through endoplasmic reticulum (ER)–mitochondria microdomain interactions, which contribute to cellular redox imbalance, mitochondrial injury, or apoptosis. From a therapeutic perspective, the molecular mechanisms of ER–mitochondria contact could help identify a therapeutic target for I/R injury and a new pertinent treatment for reperfusion damage in clinical practice [3,8,9].

On the one hand, lncRNAs comprise a heterogeneous group of RNA molecules with multiple functions and interaction partners, thus interfering with numerous endogenous signaling pathways during cardiogenesis. Long ncRNAs have different developmental stages, but they may also change their expression in response to various triggers or under physiological and pathological conditions by coding proteins or through direct binding to proteins. Therefore, lncRNAs play key functions in the occurrence of myocardial infarction and hypertrophy, heart failure, arrhythmias, and other processes that significantly influence survival in patients with cardiovascular diseases [10].

On the other hand, the most widely studied microRNAs, which are present in a multitude of cardiac cell types, are key for monitoring IR progression, and they could target transient receptor potential (TRP) channels that also participate in the pathophysiology of myocardial I/R injury [3,11,12]. Nevertheless, multiple studies are needed to improve the prevention, diagnosis, and treatment of myocardial I/R injury based on clinical efficacy for the best patient outcome.

## 2. MicroRNAs, LncRNAs, and ncRNAs in Myocardial Infarction

The difference between I/R and AMI in terms of the ncRNA profile depends on the stage of the disease, because ncRNA expression could change. Furthermore, while the ischemic area expands according to the duration and severity of blood flow reduction, maximum reperfusion is achieved in a moderate ischemic injury [4,6,7].

### 2.1. MicroRNAs in Myocardial Infarction

MicroRNAs are endogenous RNAs of ~22 nucleotides that negatively regulate the expression of target genes by usually binding to the 3′ untranslated region (UTR) of mRNAs and inhibiting their translation. They are synthesized as precursors in the nucleus, where they undergo maturation with several enzymatic reactions and are translocated to the cytoplasm, where they exert their biological function in recruiting specific silencing proteins that form the RNA-induced silencing complex (RISC) [13,14]. It has been predicted that in humans, about 60% of mRNAs are targets for miRNAs, and one miRNA may target more than 100 mRNAs [15]. Specific miRNAs are differently expressed in cardiac tissue and vascular cells, playing an important role as regulators of biological functions that include cell differentiation, growth, apoptosis, proliferation, angiogenesis, and contractility [16].

The aberrant expression of miRNAs has been reported in myocardial infarction and end-stage cardiomyopathy [17,18]. In addition, several miRNAs such as miRNA-1, miRNA-133a, miRNA-20a/b, and miRNA-499 are considered specific signaling molecules generously expressed in the myocardium [19]. Moreover, in experimental studies, the expression of these myomiRs has also been associated with arrhythmias, cardiac hypertrophy, fibrosis, and myocardial infarction [20,21].

The dysregulation of specific miRNAs in cardiac tissue has been revealed in patients with myocardial infarction, while other miRNAs such as miRNA-21-5p and miRNA-126-3p, which are not cardiac-specific or muscle-enriched molecules, contribute to the onset and progression of CVDs [22,23].

The miRNome includes miRNA-1, miRNA-20a, miRNA-21, miRNA-126, miRNA-155, miRNA-210, and miRNA-214 in myocardial infarction; miRNA-1, miRNA-17-92, miRNA-106b-25, miRNA-133, miRNA-133a, and miRNA-212 in cardiac arrhythmia; miRNA-21, miRNA-29, and miRNA133 in cardiac fibrosis; and miRNA-21, miRNA-23a, miRNA-24, miRNA-21, miRNA-29, miRNA-30, miRNA-195, miRNA-210, and miRNA-499 in cardiac hypertrophy. Other miRNAs such as miRNA-143 and miRNA-145 are correlated with smooth muscle and vascular contractile function, while miRNA-33, miRNA-155, miRNA 146a, miRNA-let7a, miRNA-21, miRNA-223, and miRNA-125a are correlated with inflammatory responses [21,22,24].

### 2.2. LncRNAs in Myocardial Infarction

Hundreds of lncRNAs have been shown to play critical roles in cardiovascular diseases, particularly in acute myocardial infarction, which is the main topic of this review article [3,25].

Myocardial hypoxia during infarction induces a great loss of viable cardiomyocytes by both necrosis and apoptosis. Recent research demonstrated the regulatory involvement of lncRNAs in apoptosis due to the infarcted heart. Besides intracellular signaling, switches in lncRNA expression levels modify intercellular dialogue by modulating paracrine communication. The aberrant expression of lncRNA myocardial infarction-associated transcript (MIAT), which is mainly expressed in the heart, was firmly associated with apoptosis, cell proliferation, and fibrotic remodeling in acute myocardial infarction [26,27].

Moreover, the knockdown of MIAT ameliorates cardiac function in the infarcted heart. MIAT also targets miR-24 and functions as its sponge in post-infarct myocardium during cardiac fibrosis. Therefore, by targeting several anti-fibrotic miRNAs such as miR-24, miR-29, miR-30, and miR-133, MIAT has shown to promote cardiac fibrosis in experimental models. Lentiviral-mediated knockdown of MIAT prior to MI reduced infarct size and interstitial fibrosis, contributing to preserved cardiac function, through the control of collagen proliferation. MIAT acts as a pro-hypertrophic lncRNA in cardiomyocytes by sponging the anti-hypertrophic miR-150, miR-93, and miR-93 via TLR4. Altogether, these findings highlight the complex regulatory network of MIAT in this cardiac disease, thus serving as an efficient therapeutic target [27].

The conserved lncRNA Wisp2 super-enhancer-associated RNA (Wisper) has been described as a powerful regulator of cardiac fibrosis in an experimental murine model of myocardial infarction (MI) as well as an attractive therapeutic target that reduces the pathological evolution of fibrosis in response to AMI, thus preventing detrimental remodeling in the impaired heart tissue [28].

In mice, treatment with GapmeR, which induces the knockdown of Wisper, a lncRNA that is up-regulated in cardiac fibroblasts after myocardial infarction, leads to reduced infarct size and fibrosis, thus preserving the cardiac structure and function [29]. Therapeutic GapmeR injections have been successfully administered to control the cardiac fibrosis (CF)-specific lncRNA maternally expressed gene 3 (Meg3) and lncRNA Wisper in MI. All these studies presented an improved cardiac function after therapeutic intervention, thus underlining the considerable potential of antisense drugs targeting lncRNAs [30,31].

In addition, secreted lncRNAs are essential for adding another layer of complexity to the role of intercellular communication by mediating cell proliferation or fibrosis progression in the area of cardiac infarction. The lncRNA MIAT1, for example, was involved in the coordination of the acute inflammatory response consecutive to myocardial infarction. In another study, it was demonstrated that oxygen-deficiency-induced up-regulation of myocardial infarction-associated transcript 1 (Mirt1) promoted fibrosis in neonatal mice and activated the nuclear transport of NF-kB and thus the expression of pro-inflammatory cytokines such as IL-6, TNF-α, and IL-1β. The accelerated flow of these pro-inflammatory cytokines in turn facilitates increased cardiomyocyte apoptosis and macrophage infiltration into the infarcted area [29].

### 2.3. Circulating ncRNAs as Biomarkers of Myocardial Infarction

Based on their genomic loci, interaction with DNA components, closeness to protein coding genes, and length, lncRNAs are classified as sense, antisense, intronic, intergenic, enhancer, bidirectional, and circular [29].

Circular RNAs displaying a circular shape (circRNAs) are a peculiar group of lncRNAs copied directly from back-spliced exons, thus producing covalently closed loop structures established by joining together the 3′ and 5′ end through circularization [32].

Circular RNAs regulate gene expression at transcriptional or post-transcriptional level, acting like a sponge and adopting circRNA–miRNA–mRNA networks. Moreover, circRNAs have higher biological stability than linear RNAs and cannot be identified or hydrolyzed by RNA exonuclease because of their circular architecture. RNA-binding motif protein 20 (RBM20), which is an important pathogenic gene of myocardial disease, is dependent on TTNcircRNAs and targets multiple key cardiac genes, such as calcium/calmodulin-dependent kinase II (CAMK2D), while also being dependent on *titin* (*TTN*) circRNAs [33]. Furthermore, certain circRNAs including *circSLC8A1*, *circSLC8A1*, *circCACNA1D*, *circSPHKAP*, and *circALPK2* are highly expressed in cardiomyocytes, even in induced pluripotent stem cells (iPSCs), and could serve as biomarkers in blood [34].

Additional circulating lncRNAs have also been defined as potential biomarkers of heart failure (smooth muscle and endothelial cell enriched migration/differentiation-associated LncRNA (SENCR), non-coding repressor of NFAT (NRON), long intergenic non-coding RNA predicting cardiac remodeling (LIPCAR), myosin heavy chain associated RNA transcript (MHRT)) and acute myocardial infarction (zinc finger antisense 1(ZFAS1), homeobox antisense intergenic RNA (HOTAIR), LIPCAR, urothelial carcinoma-associated 1 (UCA1), ANRIL, KQT-like subfamily, member 1 opposite strand/antisense transcript 1 (KCNQ1OT1), LncPPARδ, CoroMarker). One appropriate example of lncRNAs as an IMA predictor biomarker is the mitochondria-derived lncRNA LIPCAR, whose plasma levels are correlated with left ventricular remodeling after IMA and increased risk of developing heart failure.

There is a higher expression of LIPCAR in the blood of patients with successive heart failure after IMA, associated with an increased risk of cardiovascular death, compared to IMA patients without left ventricle remodeling. Similarly, circular lncRNA MIAT and SENCR were associated with left ventricular cardiac remodeling in the same patients [32,34,35]. Circulating lncRNA MHRT and NRON could be independent predictors of heart failure, and, if they are combined with elevated plasma levels of lncRNA ANRIL, of a higher risk for in-stent restenosis [36] (Table 1).

The blood expression levels of circRNA MICRA measured at reperfusion anticipate left ventricular dysfunction three to four months after IMA in two independent cohorts because of its circularization, stability in body fluids, as well as protection from endonuclease activities [37].

## 3. Ischemia–Reperfusion (I/R) Injury of the Heart

The most broadly studied ncRNAs are miRNAs, which are generously found in almost all cardiac cell types such as cardiomyocytes, endothelial cells, and fibroblasts. They control multiple cellular processes including apoptosis, cell cycle progression, proliferation, metabolism, angiogenesis, and autophagy [19]. Moreover, the dysregulation of miRNA expression utilizing inhibitors (antagomiRs) and miRNAs mimics in vitro cell lines, while in genetically modified experimental models it has a significant impact on cardiomyogenesis, thus modulating cardiac hypertrophy or fibrosis, infarct size, and intercellular communication [3,59,60,61].

Ischemia–reperfusion (I/R) injury of the heart is the main CVD correlated with coronary ischemic heart disease, acute myocardial infarction (AMI), cardiac surgery, and transplantation. Heart tissue ischemia is caused by obstruction in the coronary artery secondary to the reduction or stopping of blood flow, which determines inadequate metabolic and oxygen supply [62].

The myocardial infarct size and the long-term prognosis of heart disease depend on the intensity and duration of ischemic apoptosis triggers, as well as on the presence/absence of cardioprotective interventions. Inflammation and oxidative stress, which have been suggested to play an essential role in the evolution of I/R, are major mechanisms leading to altered homeostasis. In addition, it is largely accepted that there are other important mechanisms such as the activation of matrix metalloproteinases, apoptosis, and gene expression variation [63,64,65,66].

In contrast, reperfusion represents the recovery of post-ischemic blood flow levels, which, if controlled too late, could even intensify ischemic damage [62]. Therefore, reperfusion settings require the restoration of coronary permeability as soon as possible using angioplasty and thrombolysis, but, paradoxically, after blood flow restoration, an immune response expands the myocardial injury, thus worsening the patient’s prognosis [67,68].

The extensive loss of cardiomyocytes following acute infarction overwhelms the very limited regenerative capacity of the myocardium, giving rise to a collagen-based scar. Cardiac necrotic cells discharge endogenous danger signals, thus stimulating the innate immune pathways and leading to myocardial ischemia reperfusion syndrome (MIRS), which is a severe inflammatory response [68,69,70]. The activation of complement system and toll-like receptor (TLR) signaling contribute to the increased production of pro-inflammatory cytokines, including interleukin-1 (IL-1) and tumor necrosis factor-α (TNF-α), as well as to the production of chemokines such as monocyte chemoattractant protein-1 (MCP-1/CCL2) [71,72]. Inflammatory signals induce strong adhesive leukocyte–endothelial interactions, leading to monocyte-dependent neutrophil extravasation, which clears the infarct from dead cells. Anti-inflammatory mediators are released, and the reparative cells are activated. Therefore, infarct fibroblasts proliferate, go through transforming growth factor (TGF-β1)-mediated myofibroblast transdifferentiation, and deposit generous amounts of extracellular matrix proteins which contribute significantly to the architecture of the infarcted ventricle. Moreover, the cardioprotective effect of inflammasome activation in myocardial ischemia–reperfusion syndrome (MIRS) and the therapeutics targeting the reparative response could prevent progressive post-infarction heart failure [69,73,74] (Figure 1).

The death of cardiomyocytes and endothelial cells is the major trigger for myocardial ischemia–reperfusion syndrome (MIRS). Their discharge into the extracellular space acts as damage-associated molecular patterns (DAMPs) and comprises ATP, Ca^+^, high-mobility group box 1 protein (HMBGB1), and toxic fragments of mitochondrial DNA. The generation of DAMPs is the signal for the stimulation of the TLR9 and NLRP3 inflammasome pathways, serving as a platform for the activation of the cysteine protease caspase-1, thus inducing pyroptosis—a highly inflammatory form of nearby cardiomyocyte death, identified as a series of features that are typical of both necrosis and apoptosis. This inflammatory message also converges on the stimulation of nuclear factor-*κ*B (NF-*κ*B) and the myeloid differentiation primary response gene 88 (MyD88) pathways, amplifying the release of a great number of inflammatory mediators, including interleukin-6 (IL-6), monocyte chemoattractant protein-1 (MCP1), and tumor-necrosis factor-*α* (TNF-*α*), as well as the proinflammatory cytokines interleukin-1β (IL-1β) and IL-18, both promoting caspase-1-dependent cell death.

Recent research revealed that there are consistent changes in various groups of non-coding RNAs (ncRNAs) including miRs, long non-coding RNAs (lncRNAs) [2], and circular RNAs (circRNAs) associated with cardiac infarction, thus indicating that these molecules may exacerbate or attenuate myocardial damage due to I/R [4,5,6,7].

## 4. Role and Mechanisms of miRNAs in Ischemia–Reperfusion (I/R) Injury of the Heart

Each miR can control several targets and more than one miR can influence a single mRNA. For instance, the significant dysregulation of miR-1 and miR-21 has been associated with cardiac injury/remodeling and cardioprotection [62]. Cardiomyocite death is modulated through miR-199, miR-15b, miR-21, miR-30b, miR-34a, miR-497, and miR-1 by suppressing B-cell lymphoma 2 (BCL-2), which has an anti-apoptotic effect. The under-expression of miR-1 alleviates cardiac injury via protein kinase C (pkC) and HSP60 [3].

Myo-miR-1, acting as a specific regulator of somatic and cardiac muscle progenitors through the Notch 1 receptor, is connected with the expression of many cardiac transcription factors which are considered targets of its gene: Nkx2.5 expressed in early cardiac cells, myocardin, a specific transcription co-factor that potently induces miR-1, serum response factor (SRF) downregulating miR-1, WNT, and fibroblast growth *factor* (FGF) signaling components [8,62].

Fibronectin, Ras GTP-ase-activating protein (RasGAP), cyclin-dependent kinase-9 (Cdk9), and Ras homolog enriched in brain (Rheb) are alternative growth-related targets inhibited by the overexpression of myo-miR-1. MiR1 overexpression encourages myogenesis through the repression of histone deacetylase 4 (HDAC4) and interferes with cardiac proliferation through Hand2 transcription factor involved in myocyte expansion. The Over-expression of miR-1 induces cardiomyogenesis via suppressing FGF and WNT signaling pathways. Sox6 also efficiently drives human cardiac cell proliferation in miR-1 knockdown. During post-myocardial infarction (MI) inflammation, miR-1 suppresses the anti-apoptotic genes by targeting heat shock proteins (HSPs) such as Hsp 60, Hsp-70, *insulin-like growth factor 1* (IGF-1), and B-cell lymphoma 2 (BCL-2), all of which act as sensors to extracellular damage-associated molecular patterns (DAMPs) released from the injured cardiac tissue [75,76,77].

By contrast, miR-133, which is derived from the same miRNA polycistron as miR-1, has a specific anti-apoptotic role in amplifying myoblast proliferation and differentiation in cultured myoblasts by repressing SRFs that participate in transcriptional circuits positively controlling cardiac growth and HF [78]. Both in human patients and in experimental models, these two mature miRNAs, miR-1 and miR-133, were down-regulated in the atria and left ventricle during cardiac hypertrophy [3,78]. Moreover, antagomiR-133 preserves cardiac hypertrophy by the post-transcriptional processing of RhoA GTP-ase protein, a novel potential target for therapeutic intervention [79]. MiR-133, which blocks the pro-apoptotic genes via caspase-9 protein, is under-expressed in the myocardial zone. MiR-1 and miR-133 are down-regulated after 30 min of ischemia and 120 min after reperfusion [3].

MiR-21 expression is altered in the heart muscle and coronary artery disease if CVD conditions such as cardiac hypertrophy/fibrosis, heart failure, ischemic heart disease, and proliferative vascular disease are present [80]. Furthermore, the extracellular signal-regulated kinase 1/2 (ERK1/2)–mitogen-activated protein kinase (MAPK) cascade, the central signaling pathway that monitors apoptosis and stress response through the down-regulation of Sprouty (Spry 1) and up-regulation of matrix metalloproteinase-2 (MMP-2) genes, is activated by miR-21. A paracrine-signaling mediator, miR-21-3p, is involved in intercellular communication and transverse aortic constriction (TAC)/angiotensin II (AngII) hypertrophy [62].

MiR-21 attenuates I/R injury by targeting certain pro-survival/pro-apoptotic genes such as Fas ligand (FasL), programmed cell death (PDCD4), and phosphatase and tensin homolog (PTEN), a multi-functional tumor suppressor that activates the AKT kinase or protein kinase B signaling pathway. The inhibition of the ischemia-induced up-regulation of FasL and PTEN increasing phospho-AKT limits infarct size and heart failure (HF). MiR-21 is induced in cardiomyocites in the early phase of MI (protection) and in fibroblasts (fibrosis, cardiac remodeling) in the late phase [3]. In experimental models, the inhibition of miR-21 or antimiR-21 decreases I/R-induced adverse myocardial remodeling and the loss of ventricular myocytes, thus improving regional contractility and interfering with the inflammatory and immune response through the repression of the myocardial extracellular signal-related kinase (ERK)–MAPK signaling pathway [81].

The prohibitin complexes PHB1 and PHB2, which are overexpressed in the mitochondrial inner membrane, control the reduction in apoptosis and myocardial infarct size, the equilibrium of mitochondrial dynamics in the heart during I/R injury being extremely important. Moreover, miR-539 down-regulates the PHB2 complex, which abolishes myocardial apoptosis, improving MI [3,48].

According to current research, miR-155 could intensify the inflammatory status and thus I/R injury via the induction of TNF-α, CD105, IL-1β, and caspase 3. Moreover, miR-155 inhibition is sharply associated with apoptosis, which may protect the myocardium structure and lower the infarct area in IR injury by targeting BAG family molecular chaperone regulator 5 (BAG5), inhibiting the MAPK/JNK pathway and inflammation through the suppression of SOCS-1 [3,82].

Other studies have shown that miR-210 has a proangiogenic effect through the ephrin-3 pathway in response to ischemia and a cardioprotective effect via the *p53*-*AKT* network, monitoring mitochondrial ROS production and calcium overload. Similar to miR-210, miR-126 is a protective proangiogenic miRNA which induces vascular endothelial growth factor (VEGF) and suppresses the vascular cell-adhesion molecule (VCAM)-1 and angiopoietin-1 (Ang-1) [3]. The ERBB receptor feedback inhibitor (ERRFI)-1 is a direct target of miR-126, which could represent a potential treatment for myocardial infarction by attenuating apoptosis and ROS accumulation in I/R injury [83].

In addition, the inhibition of miR-92a by a systemic infusion of anti-miR-92a and MRG-110 derepresses miR-92a targets, thus showing a noticeable improvement in ischemia–reperfusion damage. In addition to diminishing infarct size and having a protective effect in MI, this inhibition has shown an amelioration in the peripheral blood compartment and a reduction in local inflammation, all critical processes involved in ischemia–reperfusion injury [3,84].

Furthermore, miR-144 reduces infarct size and could serve as a new potential therapeutic approach in alleviating myocardial I/R injury through the inhibition of the pro-apoptotic PTEN factor, proved as its direct target, and the consecutive activation of the phosphatidylinositol-3-kinase/ protein kinase B (PI3K/AKT) pathway [85,86]. The infarct area is also reduced through a new signaling pathway, the miR-346/Bcl-2-associated X protein (Bax) axis, which modulates cardiac apoptosis [87]. In addition, the down-regulation of miR-320, miR-214, and miR-499 has protective effects in myocardial I/R injury [88,89,90] (Table 2).

## 5. Role and Mechanisms of lncRNAs in Ischemia–Reperfusion (I/R) Injury of the Heart

Acute myocardial infarction (AMI) is defined by myocardial necrosis and the stimulation of the inflammatory response. Reperfusion therapy after AMI could adequately restore the blood supply and metabolic support of the ischemic myocardium and recover the dying myocardium. Nonetheless, myocardial I/R damage has grown into a new threat to reperfusion therapy for AMI.

In addition, the mitochondria-associated ER membrane (MAM), known as the ER–mitochondria microdomains, modulates the mitochondria/ER function and spatial structure. ER–mitochondria microdomains are crucial for cellular contraction and mobility, energy production, and physiological extracellular signal transmission. In I/R injury, MAM participates in mitochondrial injury, cellular redox imbalance, ER stress, energy depletion, and apoptosis through different mechanisms interfering with the mitochondrial integrity [8]. For instance, the blockage of mitochondrial fission through mitochondrial calcium uniporter (MCU) inhibition represses caspase activation and reduces calcium imbalance and cellular ROS generation [8].

The activation of MAM-dependent mitophagy during AMI requires the Becn1–Vps34–ATG14 complex, which represents a new proangiogenic therapeutic agent. The up-regulation of the ATG14 gene through microRNA-130a inhibition reduces cardiomyocyte apoptosis [52].

Numerous long-chain non-coding RNAs (lncRNAs) are dysregulated by I/R injury. Most of these lncRNAs control cell death in myocardial I/R injury by sponging a certain miRNA that regulates complementary signaling pathways [52,91].

Functionally, necrosis-related factor (NRF) is known as a lncRNA activated by the p53 transcription factor. It decreases miR-873 expression and depresses receptor-interacting serine/threonine-protein kinase 3 (RIPK)-1/RIPK-3-mediated necrosis of cardiomyocytes, thus reducing the myocardial infarct extent in experimental I/R injury through the P53–NRF–miR-873–RIPK1/RIPK3 axis [38,39].

Among the most markedly increased lncRNAs following MI/R, lncRNA autophagy-promoting factor (APF), which is up-regulated in mice, abrogates the inhibitory effect of ATG7 protein expression by targeting miR 188 3p [40,41].

Long-chain ncRNA cardiac autophagy inhibitory factor (CAIF) alleviates MI and has cardioprotective effects by interacting with p53, which blocks myocardin transcription, thus decreasing microtubule-associated protein 1A/1B-light chain 3 (LC3-II) accumulation and subsequent cardiac autophagy [3,39,42,43]. Long-chain ncRNA antisense transcript of beta-secretase-1 (BACE1)-AS positively activates the accumulation of toxic beta-amyloid protein in cardiomyocytes and endothelial cells, this pathway facilitating HF pathogenesis [39,47]. Cardiac apoptosis-related lncRNA (CARL) acts as a sponge of miR-539, derepressing the expression of PHB2, which is a mitochondrial apoptosis and fission inhibitor, thus improving cardiomyocyte survival [39,48].

Another example of cardiac-specific lncRNA and a new biomarker of acute MI is lncRNA zinc finger antisense 1 (ZFAS1), which modulates Ca^2+^ homeostasis—a key determinant of cardiac contractile function. The up-regulation of *ZFAS1* leads to the impairment of cardiac function, as observed in murine MI, so anti-*ZFAS1* could be considered a new therapeutic target for protecting SERCA2a activity in the ischemic myocardium [53,54]. In addition, the overexpression of lnc ZFAS1 during MI promotes myocardial cell death via the invigoration of C-reactive protein (CRP) and repression of miR-150 expression [52,55].

Long-chain ncRNAs, including the KCNQ1OT1, and lncRNA regulator of reprogramming (ROR) are up-regulated in hypoxia–reperfusion treated cardiomyocytes and I/R injury, inducing pro-apoptotic gene activation via the p38 mitogen-activated protein kinase (MAPK) and nuclear factor-kB (NF-kB) pathways [52,56,57].

The lncRNA long intergenic non-coding RNA predicting cardiac remodeling (LIPCAR) is a mitochondrial lncRNA that is decreased early during the initial phase of MI, but up-regulated during the final stages. Circulating LIPCAR was used as a biomarker to screen and monitor MI, and as a prognostic tool for cardiac failure and remodeling [3,44,45,46].

The lncRNA homeobox antisense intergenic RNA (HOTAIR), a cardioprotective lncRNA, is decreased in AMI patients and its level is inversely correlated with miR-1 and high-sensitivity cardiac troponin I (cTnI) concentration [92]. HOTAIR interacts with miR-19 and functionally blocks it. Therefore, the overexpression of HOTAIR improves the expression of PTEN (a direct target of miR-19), which is involved in the HOTAIR-mediated inhibition of cardiac hypertrophy [3,49,50]. Moreover, lncHOTAIR prevents and alleviates myocardial injury and inflammation caused by I/R by sponging miR-519d-3p [51].

The lncRNA highly up-regulated in liver cancer (HULC) has a protective effect against myocardial I/R injury, relying on the inhibition of the NLRP3/caspase-1/IL-1β signaling pathway, which is a target axis of miR-377-5p [92].

Therefore, according to the route of action, ncRNA mechanisms of action are specific to each type. miRNAs destabilize target mRNA, thus suppressing protein translation and silencing gene expression by RNA interference. In myocardial infarction, miR-294, miR-133, miR-410, miR-539, miR495, and miR-433 are over-expressed, while miR-195, miR-15, miR-497, miR-199a-3p, miR-590-3p, and miR-133 are under-expressed. The circulating miRNAs associated with AMI are miR-1254, miR-150-3p, miR-499, miR-34a, and miR-30a-5p, which are up-regulated, and miR-132-5p, which is down-regulated [58,93].

Most long non-coding RNAs and circRNAs sponge miRNAs and proteins, control chromatin changes by remodeling complexes, function as protein decoys, have a scaffold role for protein complexes, and are transcriptional enhancers for target genes [93,94,95,96,97].

Long ncRNA MIAT was identified in a case-control association study including 3435 patients with myocardial infarction versus 3774 healthy controls. This comparison highlighted six single-nucleotide polymorphisms in the MIAT locus that are strongly linked to a higher risk of AMI. Interestingly, MIAT levels measured in peripheral blood cells were different between patients with ST-segment–elevation (lower MIAT) and patients with non–ST-segment–elevation myocardial infarction. Besides MIAT, the same association was shown for lncRNAs ANRIL, MALAT1, and KCNQ1OT1 [93,94,95,96,97,98].

Another study revealed that thirty lncRNAs were either down- or up-regulated. In addition, a time-course analysis of Mirt1 and Mirt2, which are the two most strongly upregulated lncRNAs in myocardial infarction, indicated that their expression was highest 24 h after AMI and returned to normal after two days. Therefore, the study demonstrated that changes in lncRNA expression can occur rapidly and are stage-dependent. The majority of deregulated lncRNA transcripts are associated with an active, cardiac-specific enhancer [44,99].

Other clinical trials focused on the remodeling biomarker lncRNA LIPCAR, which is differentially expressed in the plasma of patients with severe left ventricular remodeling after AMI. Moreover, LIPCAR expression comes from the mitochondrial genome, revealing that lncRNAs can also originate from extranuclear DNA. Recently, lncRNA UCA1 was considered a novel biomarker of AMI. UCA1 plasma levels were significantly lower 12 h after AMI but elevated after 72 h in patients compared to controls, although its predictive power is lower than that of creatine kinase and troponin I. However, UCA1 is a pertinent example that reveals the possibility of developing a lncRNA diagnostic profile with high clinical relevance, especially since it can be detected directly through non-invasive urine tests with high specificity and sensitivity but low costs [44,100,101].

## 6. Conclusions

In conclusion, ncRNAs are ubiquitous RNA molecules that play a key role in modulating the molecular mechanisms underlying the pathogenesis of cardiovascular diseases. Both lncRNAs and miRNAs play critical functions in the pathogenesis of acute myocardial I/R injury. Accumulating evidence demonstrates that ncRNAs function as pro- or anti-MI inflammation factors through their effect on myocardial cell death and cardiomyocyte regeneration signaling pathways. This review supports the presence of cross-talk between miRNAs and lncRNAs that controls specific molecular events during MIRS. The personalized modulation of ncRNAs could be a novel therapeutic strategy to combat various cardiac disorders, including myocardial infarction. Further research is needed in order to develop a more specific and non-toxic ncRNA strand that can be used in clinical settings. New strategies for targeting ncRNAs can be beneficial in MI prevention and secondary prevention after acute MI, thus improving clinical outcomes and reducing mortality.

## Figures and Tables

**Figure 1 ijms-23-02728-f001:**
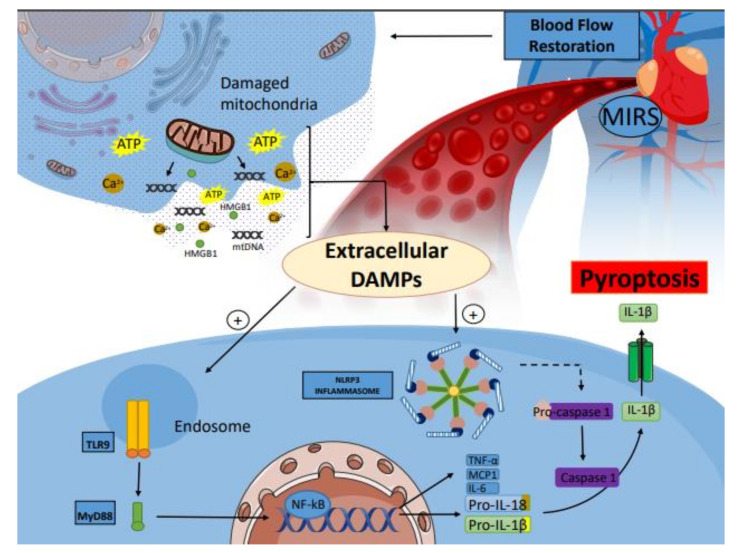
**Inflammasome activation in MIRS. Abbreviations: DAMPs**—damage-associated molecular patterns; **HMBGB1**—high-mobility group box 1 protein; **IL-1β**—interleukin-1β; **MyD88**—myeloid differentiation primary response gene 88; **MIRS**—myocardial ischemia–reperfusion syndrome; **NF-κB**—nuclear factor-κB; **TLR**—toll-like receptors; **NLRP3**—NACHT, LRR, and PYD domain-containing protein 3.

**Table 1 ijms-23-02728-t001:** Long non-coding RNAs and their function in myocardial ischemia–reperfusion (I/R) injury.

LncRNA	Expression	Functional Role	Molecular Targets	References
NRF	Up-regulated	Necrotic cardiomyocyte death	RIPK1/RIPK3, miR-873	[3,38,39]
APF	Up-regulated	miR-188-3p suppression (adaptive cell autophagy)	miR-188-3p, ATG7	[3,39,40,41]
CAIF	Up-regulated	Cardiacautophagy suppression	P53, LC3-II	[3,39,42,43]
LIPCAR	Up-regulated	MI biomarker and prognostic tool	Mitochondria	[3,44,45,46]
BACE1-AS	Up-regulated	Toxic beta-amyloid accumulation in cardiomyocytes and endothelial cells	BACE1	[39,47]
CARL	Up-regulated	Mitochondrial apoptosis and fission	miR-539, PHB2	[39,48]
HOTAIR	Down-regulated	Myocardial apoptosis IM biomarker	miR-1, miR-19, miR-519d-3p	[3,49,50,51]
ZFAS1	Up-regulated	Myocardial cell death	miR-150; CRP; SERCA2a	[52,53,54,55]
ROR	Up-regulated	Myocardial apoptosis	MAPK	[52,56]
KCNQ1OT1	Up-regulated	Myocardial apoptosis	NF-kB	[52,57]
HULC	Down-regulated	Apoptosis, inflammation	miR-377-5p	[58]

Abbreviations: APF—autophagy promoting factor; ATG7—autophagy-related protein 7; BACE1—beta-secretase-1; BACE1-AS—antisense transcript of beta-secretase-1; CAIF—cardiac autophagy inhibitory factor; CARL—cardiac apoptosis-related lncRNA; CRP—C-reactive protein; HOTAIR—homeobox antisense intergenic RNA; HULC—highly up-regulated in liver cancer; KCNQ1OT1—KQT-like subfamily, member 1 opposite strand/antisense transcript 1; LC3-II—microtubule-associated protein 1A/1B light chain 3; LIPCAR—long intergenic non-coding RNA predicting cardiac remodeling; MAPK—mitogen-activated protein kinase; NF-kB—nuclear factor kB; NRF—necrosis-related factor; PHB2—prohibitin 2; RIPK1/RIPK3—receptor-interacting protein kinases; ROR—regulator of reprogramming; SERCA2a—sarcoplasmic reticulum Ca^2+^ ATPase 2a; ZFAS1—zinc finger antisense 1.

**Table 2 ijms-23-02728-t002:** MicroRNAs and their function in myocardial ischemia–reperfusion (I/R) injury.

MicroRNA	Expression after I/R	Functional Role	Molecular Targets	References
miR-1	Down-regulated	Apoptosis	Bcl-2 HSP60, PKC	[8,62]
miR-133	Down-regulated	Apoptosis		[3]
miR-21	Down-regulated	Cell survival	PDCD4 PTEN, FasL	[3]
miR-155	Up-regulated	Inflammation, apoptosis	TNF-α, IL-1β, CD105; Caspase3, SOCS-1; BAG5, MAPK/JNK	[3,82]
miR-320	Up-regulated	Infarction, apoptosis	HSP60, Nrf-2	[3,89]
miR-214	Up-regulated	Ca 2+overload, apoptosis	Ncx1, PTEN, Bim1	[88]
miR-494			ROCK1, Caspase3, CaMKIIδ	
miR-210	Up-regulated	ROS production, Angiogenesis, Apoptosis, Ca 2+overload	AIFM3, Efna3, Ptp1b	[3]
miR-20a		Angiogenesis	Egnin3/PHD3 PTEN	
miR-126	Up-regulated	Angiogenesis, Apoptosis	ERRFI1, VEGF, Spred-1, VCAM-1, Ang-1, CXCL12	[3,83]
miR-92a	Up-regulated	Angiogenesis, Apoptosis	SIRT1, KLF2/4 ZEB2	[3,84]
miR-144	Down-regulated	Apoptosis	PTEN/AKT, FOXO1	[85,86]
miR-499	Up-regulated	Apoptosis	SOX6	[90]
miR-483	Up-regulated	Apoptosis	*MDM4*/*p53 pathway*	[88]
miR-346	Up-regulated	Apoptosis	Bax	[87]

Abbreviations: AKT—protein kinase B; Ang-1—angiopoietin-1; BAG5—BAG family molecular chaperone regulator 5; Bcl-2—B-cell lymphoma 2; CXCL12—chemokine CXC ligand 12; ERRFI1—ERBB receptor feedback inhibitor 1; FOXO1—forkhead box protein O1; HSP60—heat shock protein 60; KLF2—Kruppel-like factor 2; Ncx1–Na^+^/Ca^2+^ exchanger; Nrf-2—nuclear factor erythroid 2-related factor 2; MAPK/JNK—mitogen-activated protein kinase/Jun N-terminal kinase pathway; *MDM4* (MDMX)—murine double minute 4, a critical negative regulator of p53; PDCD4—programmed cell death 4; PI3K—phosphatidylinositol-3-kinase; PTEN—phosphatase and tensin homolog (tumor suppressor); SIRT1—sirtuin1; SOCS-1—suppressor of cytokine signaling 1; SOX6—SRY-box transcription factor 6; Spred-1—sporty-related protein; VCAM-1—vascular cell-adhesion molecule 1; VEGF—vascular endothelial growth factor; ZEB2—zinc finger E-box binding protein 1.

## Data Availability

Not applicable.

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
