# Peer review of "Non-Coding RNAs: Prevention, Diagnosis, and Treatment in Myocardial Ischemia–Reperfusion Injury"

_ijms, 2022, doi:10.3390/ijms23052728_

Round 1

Reviewer 1 Report

Recent knowledge concerning the role of non-codingRNAs (ncRNAs) in myocardial ischemia/reperfusion (I/R) injury, cardiac hypertrophy and remodeling provides new insight on their possible roles as biomarkers of cardiovascular disease (CVD), useful for early diagnosis, prognosis, as well as for treatment. MicroRNAs (miRNAs) have fewer than 200 nucleotides, while long ncRNAs (lncRNAs) have more than 200 nucleotides. LncRNAs can also present circular RNAs (circRNAs), the three types of ncRNAs, miRNAs, lncRNAs and circRNAs, acting as signaling molecules strongly involved in cardiovascular disorders. I/R injury of the heart is the main CVD correlated with acute myocardial infarction (AMI), as well as with cardiac surgery and transplantation. The expression levels of many ncRNAs and miRNAs are highly modified in the plasma of MI patients, thus having the potential to diagnose and treat MI. Cardiomyocyte and endothelial cell death is the major trigger for myocardial ischemia reperfusion syndrome (MIRS). The cardioprotective effect of inflammasome activation in MIRS and also the therapeutics targeting the reparative response could prevent progressive post-infarction heart failure. Moreover, the pharmacological and genetic modulation of these ncRNAs has the therapeutic potential to improve clinical outcomes in AMI patient. While this work is interesting, a number of concerns remain.

  1. The review topics (title) is way too broad. The authors should narrow down to I/R injury only.
  2. Some type of ncRNA array data (duplicated from published work) should be presented in I/R injury.
  3. Is there any major difference between I/R and AMI in term of ncRNA profile?
  4. The latter half of the first introductory paragraph should focus on I/R injury and current understanding for mechanism of action (consider citing some of the recent reports on this topic PMID 29962971; 26477459; 35181472)
  5. Linker RNA governing mitochondrial integrity should be highlighted. Section 5 – mechanism of action should be divided according to route of action.

6. Any clinical trials on ncRNA? Or at least pharmacotherapy targeting ncRNA.      

Author Response

Rebuttal letter to Reviewer 1

Non-Coding RNAs. Prevention, Diagnosis and Treatment in Myocardial Ischemia-Reperfusion Injury

Recent knowledge concerning the role of non-codingRNAs (ncRNAs) in myocardial ischemia/reperfusion (I/R) injury, cardiac hypertrophy and remodeling provides new insight on their possible roles as biomarkers of cardiovascular disease (CVD), useful for early diagnosis, prognosis, as well as for treatment. MicroRNAs (miRNAs) have fewer than 200 nucleotides, while long ncRNAs (lncRNAs) have more than 200 nucleotides. LncRNAs can also present circular RNAs (circRNAs), the three types of ncRNAs, miRNAs, lncRNAs and circRNAs, acting as signaling molecules strongly involved in cardiovascular disorders. I/R injury of the heart is the main CVD correlated with acute myocardial infarction (AMI), as well as with cardiac surgery and transplantation. The expression levels of many ncRNAs and miRNAs are highly modified in the plasma of MI patients, thus having the potential to diagnose and treat MI. Cardiomyocyte and endothelial cell death is the major trigger for myocardial ischemia reperfusion syndrome (MIRS). The cardioprotective effect of inflammasome activation in MIRS and also the therapeutics targeting the reparative response could prevent progressive post-infarction heart failure. Moreover, the pharmacological and genetic modulation of these ncRNAs has the therapeutic potential to improve clinical outcomes in AMI patient. While this work is interesting, a number of concerns remain.

Thank you for your useful feedback. We revised the manuscript, and we hope that the text is now clearer and more pertinent.

  1. The review topics (title) is way too broad. The authors should narrow down to I/R injury only.

We focused the manuscript on ischemia-reperfusion injury and changed the title to Non-Coding RNAs. Prevention, Diagnosis and Treatment in Myocardial Ischemia-Reperfusion Injury.

  1. Some type of ncRNA array data (duplicated from published work) should be presented in I/R injury.

We replaced and updated the references with focus on I/R injury. We also changed some paragraphs, including those on the topic of CVD, which was too broad.

  1. Is there any major difference between I/R and AMI in term of ncRNA profile?

This depends on the stage. “Furthermore, while the ischemic area expands according to the duration and severity of blood flow reduction, maximum reperfusion is achieved in a moderate ischemic injury [4, 6, 7].” For example, “the lncRNA long intergenic non-coding RNA predicting cardiac remodeling (LIPCAR) is a mitochondrial lncRNA that is decreased early during the initial phase of MI, but up-regulated during the final stages. Circulating LIPCAR was used as a biomarker to screen and monitor MI, and as a prognostic tool for cardiac failure and remodeling [3, 89, 90, 91]”.

  1. The latter half of the first introductory paragraph should focus on I/R injury and current understanding for mechanism of action (consider citing some of the recent reports on this topic PMID 29962971; 26477459; 35181472).

We added the requested reports, and we rewrote the paragraphs. (Zhou H, Wang S, Hu S, Chen Y, Ren J. ER-Mitochondria Microdomains in Cardiac Ischemia-Reperfusion Injury: A Fresh Perspective. Front Physiol. 2018 Jun 15;9:755. doi: 10.3389/fphys.2018.00755. PMID: 29962971)

The epigenetic regulation of miRNAs and lncRNAs plays a key role in IR injury also through the endoplasmic reticulum (ER) -mitochondria microdomains interactions, which contribute to cellular redox imbalance, mitochondrial injury, or apoptosis. From a therapeutic perspective, the molecular mechanisms of the ER-mitochondria contact could help identify a therapeutic target for I/R injury and a new pertinent treatment for reperfusion damage in clinical practice.

  1. Linker RNA governing mitochondrial integrity should be highlighted. Section 5 – mechanism of action should be divided according to route of action.

We added the following paragraphs to section 5.

In addition, the mitochondria-associated ER membrane (MAM), known as the ER-mitochondria microdomains, modulates the mitochondria/ER function and spatial structure.  ER-mitochondria microdomains are crucial for cellular contraction and mobility, energy production and physiological extracellular signal transmission. In I/R injury, MAM participates in mitochondrial injury, cellular redox imbalance, ER stress, energy depletion and apoptosis through different mechanisms interfering with the mitochondrial integrity [8]. For instance, the blockage of mitochondrial fission through mitochondrial calcium uniporter (MCU) inhibition represses caspase activation, reduces calcium imbalance and cellular ROS generation [8].

The activation of MAM-dependent mitophagy during AMI requires the Becn1–Vps34–ATG14 complex, which represents a new proangiogenic therapeutic agent. The up-regulation of the ATG14 gene through microRNA-130a inhibition reduces cardiomyocyte apoptosis [74].

  1. Any clinical trials on ncRNA? Or at least pharmacotherapy targeting ncRNA.      

Yes. We added at least 4 clinical trials and new references.

lncRNA MIAT was identified in a case-control association study including 3435 patients with myocardial infarction versus 3774 healthy controls. This comparison highlighted six single-nucleotide polymorphisms in the MIAT locus that are strongly linked to a higher risk of AMI. Interestingly, MIAT levels measured in peripheral blood cells distinguished between patients with ST-segment–elevation (lower MIAT) and pa-tients with non–ST-segment–elevation myocardial infarction. Besides MIAT, the same association was shown for lncRNAs ANRIL, MALAT1 and KCNQ1OT1 [97-101, 102].

Another study revealed that thirty lncRNAs were either down- or up-regulated. In addition, time-course analysis of Mirt1 and Mirt2, which are the two most strongly upregulated lncRNAs in myocardial infarction, indicated that their expression was highest 24 hours after AMI and returned to normal after two days. Therefore, the study demonstrated that changes in lncRNA expression can occur rapidly and are stage de-pendent. The majority of deregulated lncRNA transcripts are associated with an ac-tive, cardiac-specific enhancer [103].

Other clinical trials focused on the remodeling biomarker lncRNA LIPCAR, which is

differentially expressed in the plasma of patients with severe left ventricular remodel-ing after AMI. Moreover, LIPCAR expression comes from the mitochondrial genome, revealing that lncRNAs can also originate from extranuclear DNA. Recently, lncRNA urothelial carcinoma-associated 1 (UCA1) was considered a novel biomarker of AMI. UCA1 plasma levels were significantly lower 12 hours after AMI but elevated after 72 hours in patients compared to controls, although its predictive power is lower than that of creatine kinase and troponin I. However, UCA1 is a pertinent example that reveals the possibility to develop a lncRNA diagnostic profile with high clinical relevance, especially since it can be detected directly through non-invasive urine tests with high specificity and sensitivity but low costs [104-106].

Submission Date

15 February 2022

Date of this review

20 Feb 2022 03:44:33

Reviewer 2 Report

This is an excellent review on non-coding RNA and cardiovascular disease. Only criticism is that some sentences (especially in the beginning of the review) were hard to follow or including discrepancies. First three hard sentences are listed bellow. I suggest authors read manuscript once again and at some points revise text to be more readable and accurate. Or review will be having editing service? Manuscript became easier to read after first pages?

Lines 15-18 require revision. Sentence is too hard to follow, Could be 2-3 separate sentences?

Lines 34-35 Sentence should be revised. It is not accurate

Lines 39-42 Sentence is not logical. First challenges and then novel approach?

Author Response

Rebuttal letter to Reviewer 2

Non-Coding RNAs. Prevention, Diagnosis and Treatment in Myocardial Ischemia-Reperfusion Injury

This is an excellent review on non-coding RNA and cardiovascular disease. Only criticism is that some sentences (especially in the beginning of the review) were hard to follow or including discrepancies. First three hard sentences are listed bellow. I suggest authors read manuscript once again and at some points revise text to be more readable and accurate. Or review will be having editing service? Manuscript became easier to read after first pages?

Thank you for your positive feedback. We revised the manuscript, and we hope that now the text is more reader friendly.

Lines 15-18 require revision. Sentence is too hard to follow, Could be 2-3 separate sentences?

We revised lines 15-18.

Recent knowledge concerning the role of non-coding RNAs (ncRNAs) in myocardial ischemia/reperfusion (I/R) injury provides new insight into their possible roles as biomarkers for early diagnosis, prognosis, and treatment.

Lines 34-35 Sentence should be revised. It is not accurate

We revised this sentence.

Myocardial ischemia-reperfusion (I/R) injury in acute myocardial infarction (AMI) is the most important cause of morbidity and mortality worldwide [1].

 Lines 39-42 Sentence is not logical. First challenges and then novel approach?

We modified the entire paragraph. Now, the last sentence is:

Therefore, new biomarkers for the prevention, monitoring and treatment of I/R must be identified and the most important challenges of such an integrative therapeutic approach must be understood.

Round 2

Reviewer 1 Report

The authors have addressed all my concerns in a fine manner